# Surveillance of *Salmonella enterica serovar* Typhi in Colombia, 2012–2015

**Paula Diaz-Guevara**[1]*, **Lucy Angeline Montaño**[1], **Carolina Duarte**[1], **Gabriela Zabaleta**[1], **Mailis Maes**[2], **Julio Cesar Martinez Angarita**[3], **Duy Pham Thanh**[4], **William León-Quevedo**[5], **Carlos Castañeda-Orjuela**[5], **Claudia Jimena Alvarez Alvarez**[3], **Jaime Guerrero**[3], **Miriam Moroni**[6], **Josefina Campos**[6], **Enrique Pérez**[7], **Stephen Baker**[2]*

**1** Grupo de Microbiología, Instituto Nacional de Salud, Bogotá, Colombia, **2** Cambridge Institute of Therapeutic Immunology & Infectious Disease (CITIID), The University of Cambridge, Cambridge, United Kingdom, **3** Vigilancia y Análisis de Riesgo en Salud Pública, Instituto Nacional de Salud, Bogotá, Colombia, **4** Oxford University Clinical Research Unit, The Hospital for Tropical Diseases, Ho Chi Minh City, Vietnam, **5** Observatorio Nacional de Salud, Instituto Nacional de Salud, Bogotá, Colombia, **6** Red Pulsenet Latino América y el Caribe, Subregional, Instituto Carlos Malbran, Buenos Aires, Argentina, **7** Health Emergencies Department, Pan American Health Organization/World Health Organization, PAHO/WHO, Washington DC, United States of America

* pdiaz@ins.gov.co (PDG); sgb47@medschl.cam.ac.uk (SB)

**Data Availability Statement:** All relevant data are within the manuscript and its Supporting Information files.

## Abstract

*Salmonella* Typhi (*S*. Typhi) is the causative agent of typhoid fever; a systemic disease affecting ~20 million people per year globally. There are little data regarding the contemporary epidemiology of typhoid in Latin America. Consequently, we aimed to describe some recent epidemiological aspects of typhoid in Colombia using cases reported to the National Public Health Surveillance System (Sivigila) between 2012 and 2015. Over the four-year reporting period there were 836 culture confirmed cases of typhoid in Colombia, with the majority (676/836; 80.1%) of reported cases originated from only seven departments. We further characterized 402 *S*. Typhi isolates with available corresponding data recovered from various departments of Colombia through antimicrobial susceptibility testing and molecular subtyping. The majority (235/402; 58.5%) of these typhoid cases occurred in males and were most commonly reported in those aged between 10 and 29 years (218/402; 54.2%); there were three (0.74%) reported fatalities. The overwhelming preponderance (339/402; 84.3%) of *S*. Typhi were susceptible to all tested antimicrobials. The most common antimicrobial to which the organisms exhibited non-susceptibility was ampicillin (30/402;7.5%), followed by nalidixic acid (23/402, 5.7%). Molecular subtyping identified substantial genetic diversity, which was well distributed across the country. Despite the diffuse pattern of *S*. Typhi genotypes, we identified various geographical hotspots of disease associated with local dominant genotypes. Notably, we found limited overlap of Colombian genotypes with organisms reported in other Latin American countries. Our work highlights a substantial burden of typhoid in Colombia, characterized by sustained transmission in some regions and limited epidemics in other departments. The disease is widely distributed across the country and associated with multiple antimicrobial susceptible genotypes that appear to be restricted to Colombia. This study provides a current perspective for typhoid in Latin

**Funding:** This study received funding from the Instituto Nacional de Salud, Bogotá, Colombia. The funders had no role in study design, data collection and analysis, decision to publish, or preparation of the manuscript.

**Competing interests:** The authors have declared that no competing interests exist.

America and highlights the importance of pathogen-specific surveillance to add insight into the limited epidemiology of typhoid in this region.

## Author summary

Typhoid fever is a systemic infectious disease of humans caused by the bacterium *Salmonella* Typhi. Typhoid fever is transmitted by contaminated food and water and is considered endemic in many low- and middle-income countries (LMICs) in Africa and Asia. In contrast, typhoid fever is less commonly reported in Latin America; therefore, we aimed to contribute to the knowledge of Typhoid fever in Colombia. Our data suggests a substantial burden of typhoid in Colombia, which is characterized by continual transmission in some regions and temporary epidemics in other locations. The disease is widely distributed throughout Colombia and associated with multiple genotypes that are largely susceptible to the majority of antibiotics used to treat the infection. It appears that the current epidemiology of typhoid in Colombia is distinct from Africa and Asia and largely restricted to organisms that are circulating nationally rather than internationally. This study provides a recent perspective for typhoid in Latin America and highlights the importance of pathogen-specific surveillance to add insight into the epidemiology of typhoid in this region.

## Introduction

*Salmonella enterica* serovar Typhi (*S.* Typhi) is the causative agent of typhoid fever, a systemic disease that occurs only in humans [1]. *S.* Typhi is transmitted through contaminated food and water or via contact with fecal material from acute or chronically infected individuals [1,2]. The annual global burden of typhoid is estimated to be 20.6 million cases with 223,000 deaths [3]. Typhoid is endemic in parts of South Asia, sub-Saharan Africa, Southeast Asia, and also Latin America [4]. Outbreaks and sporadic cases are common in many low- and middle-income countries (LMICs) within these regions, particularly in locations where sanitary conditions are poor [5].

Antimicrobial resistance (AMR) has become a major global issue in typhoid. The evolution and international spread of AMR in *S.* Typhi in Asia and Africa has been mainly driven by a clonal expansion of a specific haplotype (H58/genotype 4.3.1) [6,7]. These organisms are frequently multi-drug resistant (MDR) (resistant to ampicillin, chloramphenicol, and co-trimoxazole), and often exhibit reduced susceptibility to fluoroquinolones [8]. More recently, an extensively drug-resistance (XDR) *S.* Typhi clone carrying a plasmid encoding resistance to fluoroquinolones and third generation cephalosporins has been reported in Pakistan [9].

Typhoid is largely accepted to be endemic in parts of Latin America; it is estimated that the region has a medium incidence of typhoid fever (53/100,000 people) corresponding with >273,000 cases annually [4,5]. However, despite this estimation, the burden of disease in specific Latin American countries, the epidemiology, and the population structure of the circulating organisms are ill defined. Within the region, Colombia is considered to have a particularly low burden of typhoid fever [10], but much of the available data regarding typhoid fever in Colombia are historic and contemporary data are limited. In 2003 Colombia reactivated a national typhoid surveillance program and the notification of typhoid cases to the National Surveillance System Public Health (Sivigila) became mandatory in 2006 [11]. According to the official reporting system, the national incidence of typhoid (and paratyphoid) fever in Colombia remained relatively stable between 2008 and 2012 with a mean of 0.16 cases per 100,000

inhabitants annually [12]. In 2013, the incidence of the disease increased to 1.95 per 100,000 inhabitants and then declined to 0.16, 0.38, and 0.48 cases per 100,000 inhabitants in 2014, 2015, and 2016, respectively [13]. Routine surveillance data revealed a fluctuating trend of typhoid fever between differing departments in Colombia and raised concerns about the emergence and spread of AMR *S*. Typhi.

With the aim of assessing the geographical distribution and disease trends of typhoid fever in Colombia, we examined a collection of *S*. Typhi isolates with corresponding metadata accumulated by the national surveillance system between 2012 and 2015. Our specific objectives were to provide a more detailed insight into the distribution of typhoid fever in Colombia by characterizing *S*. Typhi organisms isolated from various Colombian departments with differing disease incidences via genotyping, antimicrobial susceptibility profiling, and assessing the geographical distribution of the cases.

## Methods

### Ethics statement

The study was conducted according to the principles expressed in the Declaration of Helsinki. Based on the policy of Instituto Nacional de Salud, Colombia, this study involved analysis of routinely collected surveillance data and thus did not require ethical review. The collection and use of clinical information or human biological specimens were conducted with prior oral informed consent from patients with suspected typhoid fever. Patients were offered diagnostic testing through the routine culture of stool and blood specimens as part of standard clinical care. Patient data was reviewed and analyzed anonymously.

### Study design

This was a retrospective study using data from the using available data from various departments of Colombia from cases reported to the National Surveillance System Public Health (Sivigila) between 2012 and 2015 with the code for typhoid and paratyphoid fever (INS320). The 836 typhoid cases were defined as those with a laboratory confirmed positive blood, stool, sterile fluids, or bone marrow culture for *Salmonella* Typhi [14]. These data were associated with a department of Colombia and a known population size to calculate the annual minimum incidence of disease using data from DANE 2019 (https://www.dane.gov.co/index.php/estadisticas-por-tema/demografia-y-poblacion/series-de-poblacionand) and INS cases report data (http://www.ins.gov.co/buscador-eventos/Paginas/Info-Evento.aspx.) (S1 Table). Due to limited data in national disease reporting systems, only a subset (402) of isolates had additional data and were available for microbiological characterization and genotyping, this subset of data is shown in S2 Table. Microbiological data were combined with anonymized social and demographic data, including sex, age, ethnic group, geographic location, and type of health coverage according to the General Social Security Health System (GSSSH). Areas with lower living standard and with sustained reporting of typhoid fever over time were considered to be endemic for this disease; other areas with incidences lower than the national average were considered to be sporadic for this disease.

### Bacterial identification and antimicrobial susceptibility testing

All isolates were identified and characterized phenotypically using standard biochemical testing (Triple Sugar Iron Agar (TSI), Citrate, Urea, and motility), and the API20E biochemical test kit (Biomerieux, USA). The Kauffmann-White-Le Minor serological scheme using specific commercial antisera was used to identify organisms suspected to be *S*. Typhi (Difco, United

States) [15]. Antimicrobial susceptibility testing was performed using the Kirby-Bauer disk diffusion method against amoxicillin-clavulanic acid (AMC), chloramphenicol (CHL), nalidixic acid (NAL), and tetracycline (TET). Minimum Inhibitory Concentrations (MIC) were determined using the MicroScan autoSCAN-4-System (Beckman Coulter) against ampicillin (AMP), cefotaxime (CTX), ceftriaxone (CRO), ceftazidime (CAZ), ciprofloxacin (CIP), and trimethoprim-sulfamethoxazole (SXT), according to the CLSI standards [16]. MDR was defined as resistance to ampicillin, chloramphenicol, and co-trimoxazole. We additionally aimed to identify potential Extended-Spectrum Beta-Lactamase (ESBLs) activity mediated by the $bla_{SHV}$, $bla_{TEM}$, and $bla_{CTX-M}$ genes by PCR amplification as previously described [17].

## Molecular subtyping

All 402 available organisms were subtyped following standardized PulseNet protocols using Pulsed Field Gel Electrophoresis (PFGE) [18]. Genomic DNA was restriction digested with *Xba*I enzyme (Promega, USA) and *Salmonella* Braenderup H9812 used as the reference standard. Digested genomic profiles were analyzed with Gelcompare 4.0 software (Applied Maths, Belgium) applying the Dice coefficient and UPGMA method; tolerance and optimization were set at 1.5% [19]. All PFGE data were uploaded to the Regional Database of the PulseNet Latin America and Caribbean Network (PNLA&C) hosted by the PAHO/WHO. All PFGE pattern codes were assigned following the PulseNet International guidelines for nomenclature, which includes two letters for the country or region, three letters for the serovar, three characters for the enzyme and four digits for the profile number (e.g. ALJPPX01.0001)[20].

## Data analysis

ArcGIS 9.3 software (ESRI, Redlands, CA, USA) was used for geographical localization of the various PFGE patterns. The scale of the base cartography information for the entire country or individual departments was 1:25,000. Geospatial clustering was assessed using a Kernel density estimation (KDE), with the estimation was applied to whole data set using quartic kernel function with a bandwidth in a scale of outputs and 1Km for Cucuta city, with an analysis of 5Km drawing a circle with radius (r) around each point pattern and dividing point number inside the circle by its area [21]. This approach permitted us to identify hotspots for typhoid, which was defined case clustering within the defined spatial distribution [22]. All maps were constructed using R software version 3.4.4. Data were compiled, tabulated, and ordered using Microsoft Excel and all statistical analyses was performed in R statistical software 3.4.4. PFGE analyses were performed to identify the most representative cluster *Xba*I-PFGE (III) grouping and to compare the distribution of the most frequent PFGE patterns (COINXXJPPX01.0115 and COINXXJPPX01.0008 against the minor PFGE patterns) with respect to the selected variable: origin of the isolate, sex of the patient, year of isolation, and antimicrobial susceptibility profile. Chi-squared test and Fisher's test was used to test the association between PFGE cluster and antimicrobial susceptibility profile. A 95% confidence interval (95CI) was calculated for all statistical variables; a *p* value of <0.05 was considered statistically significant.

## Results

### The epidemiology of typhoid fever in Colombia

Typhoid fever is a reportable disease in Colombia, and we reviewed all available data regarding typhoid in the regional Colombian reporting systems inclusively between 2012 and 2015. Over the four-year reporting period there were 836 culture confirmed cases of typhoid (S1 Table). The cases were widely distributed through the country and 27 of the 32 departments reported

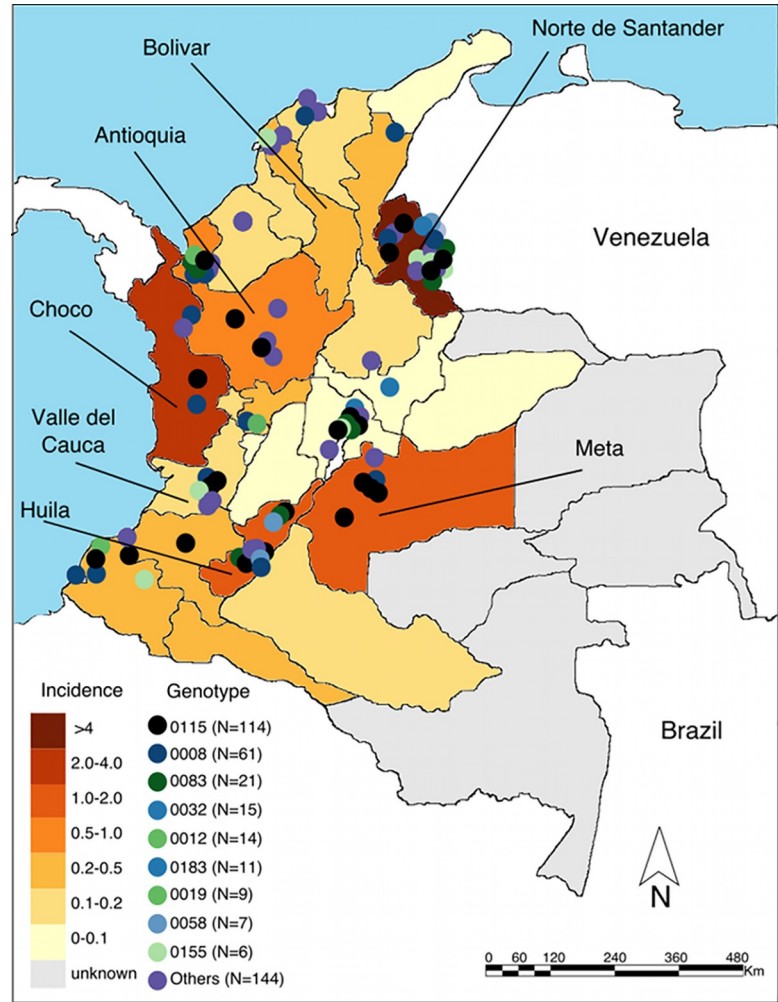

**Fig 1. The location of 402 cases of typhoid fever in Colombia (2012–2015).** Map of Colombia with a 1:25,000 scale base cartography and a Kernel function bandwidth outputs scale of 480 Km. Colored dots represents the residential location and number of typhoid fever cases throughout the country by PFGE genotype (COINJPPX01-) (see key). The seven departments with the greatest number of cases are labelled and the departments are color coded according to their mean minimum incidence per 100,000 people over the study period (S1 Table, see key). Political division map constructed in ArcGIS 9.3 software (ESRI, Redlands, CA, USA) from polygon shapefile accessed from https://www.diva-gis.org/.

the disease (Fig 1). The majority (676/836; 80.1%) of reported cases originated from only seven departments; Antioquia (*n* = 233), Norte de Santander (*n* = 233), Huila (*n* = 57), Meta (*n* = 44), Chocó (n = 41), Bolivar (n = 35), and Valle del Cauca (n = 33) (Fig 1). Using these data, we were able to calculate a mean minimum incidence of disease, which ranged from 0.02/100,000 persons per year in Tolima to 4.34/100,000 persons per year in Norte de Santander, which is in the Northeast of the country and located on the border with Venezuela (Fig 1). The combined mean minimum incidence of disease over the four-year period of surveillance was 0.44/100,000 persons per year (S1 Table).

Additional data and a paired organism were available for 402/836 (48.1%) cases in the national database (S2 Table). The preponderance (235/402; 58.5%) of these typhoid cases occurred in males and almost half of cases were affiliated to the national contributory health insurance scheme (195/402; 48.5%). Typhoid fever was most commonly reported in those

**Table 1. The demographic features of typhoid fever in Colombia, 2012–2015.**

| Variable | Category | 2012 | 2013 | 2014 | 2015 | Total |
|---|---|---|---|---|---|---|
| | | Cases (%) | Cases (%) | Cases (%) | Cases (%) | Cases (%) |
| **Sex** | Female | 21 (31.8) | 34 (39.1) | 42 (41.6) | 70 (47.3) | 167 (41.5) |
| | Male | 45 (68.2) | 53 (60.9) | 59 (58.4) | 78 (52.7) | 235 (58.5) |
| **Health coverage** [a] | Contributory | 33 (50.0) | 55 (63.2) | 38 (37.6) | 69 (46.6) | 195 (48.5) |
| | Special | 4 (6.1) | 0 (0.0) | 2 (2.0) | 7 (4.7) | 13 (3.2) |
| | No affiliation | 8 (12.1) | 9 (10.3) | 10 (9.9) | 8 (5.4) | 35 (8.7) |
| | Exception | 3 (4.5) | 2 (2.3) | 2 (2.0) | 6 (4.1) | 13 (3.2) |
| | Subsidized | 18 (27.3) | 21 (24.1) | 49 (48.5) | 58 (39.2) | 146 (36.3) |
| **Ethnic background** | Raizal | 0 (0) | 0 (0.0) | 1 (1.0) | 0 (0.0) | 1 (0.2) |
| | Afro-Colombian | 0 (0) | 5 (5.7) | 5 (5.0) | 16 (10.8) | 26 (6.5) |
| | Others | 66 (100) | 82 (94.3) | 95 (94.1) | 132 (89.2) | 375 (93.3) |
| **Age groups** | <1 year | 0 (0.0) | 0 (0.0) | 0 (0.0) | 1 (0.7) | 1 (0.2) |
| | 1 to 9 years | 8 (12.1) | 8 (9.2) | 24 (23.8) | 22 (14.9) | 62 (15.4) |
| | 10 to 1 9 years | 12 (18.2) | 22 (25.3) | 27 (26.7) | 40 (27.0) | 101 (25.1) |
| | 20 to 29 years | 24 (36.4) | 32 (36.8) | 20 (19.8) | 41(27.7) | 117 (29.1) |
| | 30 to 39 years | 12 (18.2) | 14 (16.1) | 15 (14.9) | 26 (17.6) | 67 (16.7) |
| | 40 to 49 years | 6 (9.1) | 8 (9.2) | 10 (9.9) | 6 (4.1) | 30 (7.5) |
| | >50 years | 4 (6.1) | 3 (3.4) | 5 (5.0) | 12 (8.1) | 24 (6.0) |
| **Area** [b] | Municipal center | 51 (77.3) | 72 (82.8) | 47 (46.5) | 117 (79.1) | 287 (71.4) |
| | Populated Center | 5 (7.6) | 7 (8.0) | 14 (13.9) | 10 (6.8) | 36 (9.0) |

a) Health coverage corresponds to the various healthcare schemes. Contributive: health system through which all persons linked through an employment contract, public servants, pensioners, retirees and independent workers with payment capacity make a monthly contribution to the health system. Special: social security regimes of members of the national police, military forces, navy, and air force. Exception: social security system for members of the national social benefits funds for teachers, public servants of Ecopetrol as well as servants of public universities. Subsidized: a mechanism through which the poorest population, without payment capacity, has access to health services through a subsidy offered by the state.

b) Municipal center: the geographical area defined by an urban perimeter; whose limits are established by agreements of the Municipal Council. It corresponds to the place where the administrative headquarters of a municipality is located. Populated center: A concentration of at last twenty contiguous, neighboring or semidetached houses, located in the rural area of a municipality or a Department Corregimiento.

aged ≥15 years (274/402; 68.15%) and was less frequently reported in children aged 0–4 (29/402, 7.2%) and 6–14 years (99/402, 24.6%). The organisms originated from blood (364/402; 90.5%), stool (30/402; 7.5%), sterile body fluids (2/402; 0.5%), and bone marrow (2/402; 0.5%). Approximately 90% (353/402) of these typhoid cases were hospitalized and three (0.74%) cases died (Table 1). The majority (287/402; 71.4%) of these typhoid cases originated from the municipal center of the respective departments. More specifically, the *S*. Typhi isolates were recovered from both sporadic cases (299) and notified outbreaks (103) in 15 different departments. Four of these departments (Norte de Santander (*n* = 172), Antioquia (*n* = 115), Meta (*n* = 25), and Huila (*n* = 22)) were considered to be typhoid endemic by Sivigila (Fig 1). The remaining 68 available organisms were isolated in 11 departments where typhoid was considered sporadic.

## Antimicrobial susceptibility

The overwhelming majority (339/402; 84%) of the screened *S*. Typhi organisms were susceptible to all tested antimicrobials. The remaining 63/402 (16%) isolates were non-susceptible (intermediate or resistant) to either one (*n* = 45), two (15), or more (*n* = 3) of the tested

**Table 2. The antimicrobial susceptibilities of Colombian *S.* Typhi, 2012–2015.**

| Antimicrobial | Non-susceptible; N (%) [*] |
|---|---|
| Ampicillin | 30 (7.5) |
| Amoxicillin-clavulanic acid | 3 (0.7) |
| Chloramphenicol | 3 (0.7) |
| Ciprofloxacin | 9 (2.2) |
| Nalidixic acid | 33 (5.7) |
| Trimethoprim/sulfamethoxazole | 7 (1.7) |
| Tetracycline | 12 (3.0) |

[*] 402 *S.* Typhi isolates tested

antimicrobials. The most common antimicrobial to which the organisms exhibited non-susceptibility was ampicillin (28/402; 6.9%), followed by nalidixic acid (23/402; 5.7%). Non-susceptibility against fluoroquinolones (ciprofloxacin) was uncommon (9/402; 2.2%) (Table 2). Resistance against ampicillin was more prevalent in Norte de Santander and Antioquia and the four organisms found to be MDR were isolated throughout the study period in four different regions (Bogotá, Antioquia, Norte de Santander, and Risaralda). Three organisms were found to be $bla_{TEM-1}$ positive by PCR amplification, but none were confirmed to be resistant to third generation cephalosporins.

## Genotyping of Colombian Salmonella Typhi

We pulsotyped the 402 available *S.* Typhi isolates by *Xba*I-PFGE; 113 different restriction patterns were identified (Fig 2). The estimated genetic variability in this collection of Colombian *S.* Typhi was 28.1% and the genetic similarity was 49.7%. Of the 113 individual PFGE patterns identified, nine were widely distributed throughout the country and shared between up to six departments. The most commonly identified restriction patterns were COINJPPXO1.0115 (*n* = 114), COINJPPXO1.0008 (*n* = 61), COINJPPXO1.0083 (*n* = 21) (Table 3). These three patterns represented approximately half (196/402; 48.6%) of the pulsotyped isolates and were widely distributed across the country (Fig 2 and Table 3). We investigated potential epidemiological/microbiological associations with the two most common restriction patterns (COINJPPX01.0115 and COINJPPX01.0008) but found no significant association for either pulsotype with patient age, location, sex, antimicrobial susceptibility profile, or year of isolation (S2 Table).

A clustering analysis of the *Xba*I-PFGE digestions (cutoff >75% identity) distinguished four clonal pulsotype groups that we designated group I-IV. Group I (13/402; 3.2%) consisted of 13 isolates separated into two subgroups (Ia and Ib) and were isolated in seven departments (Antioquia, Bogotá, Cesar, Huila, Meta, Norte de Santander, and Santander). Group II (27/402; 6.7%) was comprised of 27 isolates in 2 sub-groups (IIa and IIb); these organisms were again widely distributed between departments (Antioquia, Huila, Norte de Santander, Risaralda, Santander, and Valle). Group III was the prevailing (356/402; 88.5%) pulsotype group with 82 different *Xba*I-PFGE patterns falling in four sub-groups (IIIa-IIId). These isolates were identified in 14 departments (Antioquia, Atlántico, Bogotá, Bolivar, Boyacá, Caldas, Cauca, Cundinamarca, Huila, Meta, Nariño, Norte de Santander, Risaralda, and Valle) (Fig 2 and Table 3). Lastly, group IV was the smallest group (6/402; 1.5%) and were isolated in four different departments (Antioquia, Norte de Santander, Santander, and Valle), each had a different restriction pattern (Table 4).

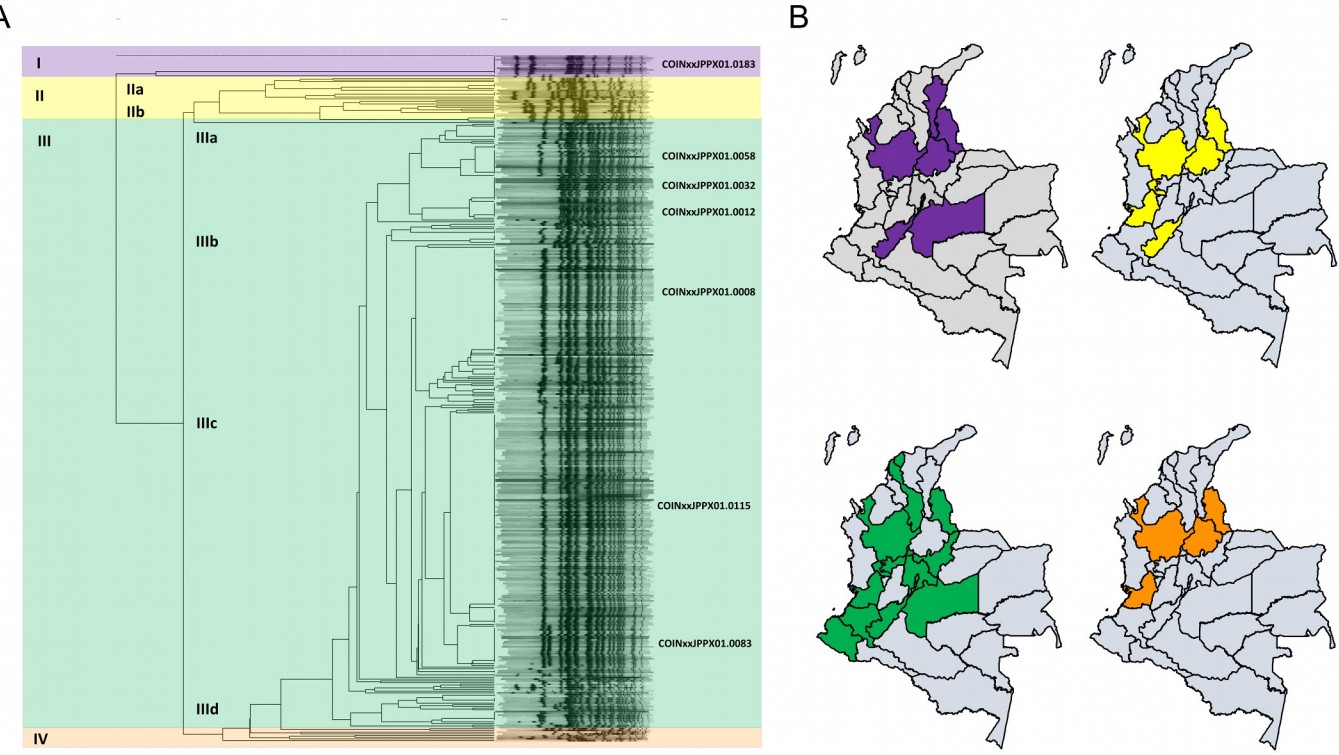

**Fig 2. The distribution of major PFGE types Colombia (2012–2015).** A) A PFGE-*Xba*I dendrogram of *S.* Typhi isolates in Colombia 2012–1015 indicating clusters I-IV (color coded); key genotypes are indicated (genetic similarity 49.7%) B) The geographical distribution of major *S.* Typhi PFGE types by Colombian department. Top left (purple); the seven departments in which cluster I (3 PFGE patterns) isolates were identified. Top right (yellow); the six departments in which cluster II (21 PFGE patterns) isolates were identified. Bottom right (green); the fourteen departments in which cluster III (82 PFGE patterns) isolates were identified. Bottom left (orange); the four departments in which cluster IV (6 PFGE patterns) isolates were identified. Political division maps constructed in R studio (VERSION) from polygon shapefiles accessed from DIVA-gis (https://www.diva-gis.org/gdata).

**Table 3.** *Salmonella* Typhi PFGE-*Xba*I patterns shared between Colombian departments.

| Location | N (%) | PFGE-Pattern; COINJPPX01- (cluster grouping) | | | | | | | | |
|---|---|---|---|---|---|---|---|---|---|---|
| | | 0008 (III) | 0012 (III) | 0023 (III) | 0032 (III) | 0058 (III) | 0115 (III) | 0155 (III) | 0156 (III) | 0183 (I) |
| Antioquia | 115 (28.6) | 49 | 13 | - | - | - | 3 | - | - | 5 |
| Atlántico | 4 (1) | - | - | - | 1 | - | - | - | - | - |
| Bogotá | 20 (5) | - | - | 1 | 2 | 2 | 4 | 1 | 1 | 1 |
| Bolivar | 6 (1.5) | 1 | - | - | - | - | 1 | - | 1 | - |
| Boyacá | 1 (0.3) | - | - | - | - | - | - | - | - | - |
| Cundinamarca | 2 (0.5) | - | - | - | - | 1 | - | - | - | - |
| Cauca | 1 (0.3) | - | - | - | - | - | - | - | - | - |
| Cesar | 1 (0.3) | - | - | - | - | - | - | - | - | 1 |
| Huila | 22 (5.5) | 1 | - | 2 | - | - | - | 5 | 1 | 1 |
| Norte de Santander | 172 (42.3) | 1 | - | - | - | - | 103 | - | - | 2 |
| Nariño | 11 (2.7) | 8 | 1 | - | - | - | - | - | - | - |
| Meta | 25 (6.2) | - | - | - | 12 | 4 | - | - | - | - |
| Risaralda | 3 (0.7) | 1 | - | - | - | - | - | - | - | - |
| Santander | 7 (1.7) | - | - | - | - | - | 3 | - | - | 1 |
| Valle | 12 (3.0) | - | - | - | - | - | - | - | - | - |
| Total | 402 | 61 | 14 | 3 | 15 | 7 | 114 | 6 | 3 | 11 |

**Table 4. The distribution of *S*. Typhi by antimicrobial susceptibility profile and relationship with PFGE cluster by Colombian department.**

| Antimicrobial susceptibility profile | N | Cluster | PFGE-*Xba*I Pattern COINJPPX01- | Departments (Year) |
|---|---|---|---|---|
| AMP(R) | 2 | I | 0183 | Antioquia (2015) |
| | 5 | III | 0008(2)-0012-0019-0085 | Antioquia (2015) |
| | 1 | III | 0058 | Bogotá (2015) |
| | 1 | III | 0032 | Meta (2015) |
| | 9 | III | 0005-0083(2)-0115(3) -0124-0171-0186 | Norte de Santander (2015) |
| AMP(I) | 1 | IV | 0110 | Norte de Santander (2015) |
| | 1 | III | 0008 | Antioquia (2015) |
| | 3 | III | 0083-0115(2) | Norte de Santander (2013–2015) |
| AMP(I)-CIP(I) | 1 | III | 0008 | Nariño (2015) |
| AMP(R)-SXT(R) | 3 | III | 0090,0096 | Antioquia (2015) |
| NAL(I)-AMP(R) | 1 | III | 0008 | Nariño (2015) |
| NAL(I)-CIP(I) | 1 | II | 0135 | Norte de Santander (2012) |
| | 1 | III | 0074 | Norte de Santander (2012) |
| | 3 | III | 0008(3) | Nariño (2015) |
| NAL(R) | 1 | III | 0032 | Meta (2015) |
| | 2 | III | 0019–0230 | Antioquia (2012–2015) |
| | 1 | III | 0127 | Bolívar (2013) |
| | 4 | III | 0006(3), 0106 | Huila (2013–2015) |
| | 1 | IV | 0221 | Antioquia (2012) |
| NAL(I) | 2 | III | 0074,0083 | Norte de Santander (2012–2015) |
| | 1 | IV | 0075 | Norte de Santander (2012) |
| NAL(R)-CIP(I) | 1 | III | 0032 | Meta (2015) |
| | 1 | III | 0187 | Norte de Santander (2014) |
| NAL(R)-CIP(I) | 1 | III | 0120 | Valle (2015) |
| SXT(R) | 1 | III | 0008 | Antioquia (2012) |
| | 1 | III | 0115 | Norte de Santander (2015) |
| | 1 | III | 0032 | Meta (2015) |
| TET(R) | 1 | II | 0224 | Norte de Santander (2014) |
| | 2 | III | 0008(2) | Antioquia (2014) |
| | 1 | III | 0156 | Bogotá (2015) |
| | 3 | III | 0115(2)-0214 | Norte de Santander (2012,2013) |
| TET(R)-AMC(I) | 1 | III | 0218 | Norte de Santander (2014) |
| TET(R)-CHL(R) | 1 | III | 0115 | Norte de Santander (2015) |
| TET(R)-CHL(R)-AMC(R) | 1 | III | 0199 | Bogotá (2012) |
| TET(R)-CHL(R)-NAL(I)-CIP(I) | 1 | II | 0193 | Risaralda (2015) |
| TET(R)-NAL(I)-AMC(I)-SXT(R)-AMP(R) | 1 | I | 0183 | Antioquia (2014) |
| Total | 63 | | | |

**Antimicrobials:** AMC; amoxicillin-clavulanic acid, AMP; ampicillin, CHL; chloramphenicol, CTX; cefotaxime, CAZ; ceftazidime, CIP; ciprofloxacin, NAL; nalidixic acid, STX; trimethoprim/Sulfamethoxazole, TET; tetracycline.

## The spatial distribution of Salmonella Typhi pulsotypes in Colombia

We further investigated the spatial distribution of the most common pulsotypes and also pulsotypes associated with outbreaks. We performed a kernel spatial analysis, mapping the most frequently identified PFGE patterns in higher incidence areas. Pulsotype COINXXJPPX01.0115 in Group III was found in five departments but was most commonly identified in Cucutá city

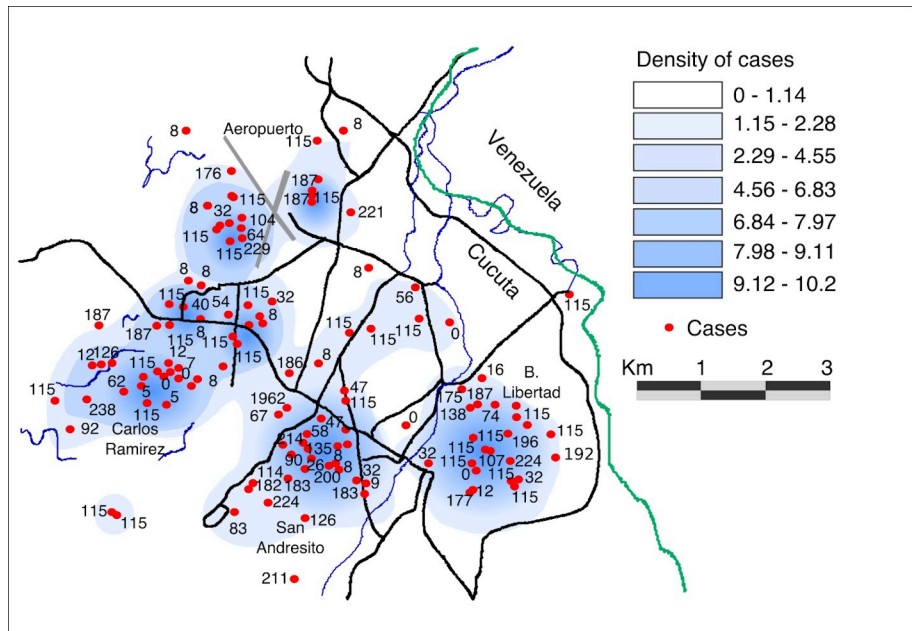

**Fig 3. Clustering of typhoid fever in Cucutá city.** 15 km Kernel density plot map of Cucutá city in Norte de Santander province bordering Venezuela showing the spatial clustering of typhoid fever cases. The darker blue shows a higher intensity of cases. Red dots highlight the individual typhoid cases and their corresponding PFGE-*Xba*I digestion patterns. Map manually constructed in ArcGIS 9.3 software (ESRI, Redlands, CA, USA).

(Norte de Santander department); and clustered in the localities of Cucutá ($n = 75$) ($X^2$ 38.9944; $p<0.0001$), Villa del Rosario ($n = 7$) ($X^2$ 12.9331; $p<0.0001$), and Los Patios ($n = 10$) ($X^2$ 2.5302; $p$ 0.11168435). This analysis also suggested a high density of cases in the neighborhoods of Aeropuerto, Libertad, San Andresito, and Carlos Ramirez in the south and west of Cucutá city (Fig 3). Pulsotype COINXXJPPX01.0008 was also in Group III and identified in six departments but clustered in Loma Verde village, in Apartadó (Antioquia department) ($n = 39$) ($X^2$ 10.4580; $p<0.0001$) (Fig 4). The next most common pulsotypes were COINXXJPPX01.0083, COINXXJPPX01.0032, and COINXXJPPX01.0155; these pulsotypes clustered in Norte de Santander department ($n = 21$) ($X^2$ 116.4260; $p<0.0001$), Meta department ($n = 12$) ($X^2$ 43.1324; $p<0.0001$), and Huila department ($n = 5$) ($X^2$ 16.4014; $p<0.0001$); respectively (Fig 4).

Pulsotypes associated with reported outbreaks (103 isolates) again typically belonged to group III. Between January and April 2012, four typhoid cases were cultured confirmed in Huila department and shared the same COINJPPXO1.0155 restriction pattern. The local water supply was postulated to be the most probable source. Further, between March and June 2015, 15 confirmed *S.* Typhi cases were reported in Cucutá (Norte de Santander department). These isolates displayed several PFGE patterns; however, 5/15 (33.3%) were identical (COIN-JPPXO1.0083). The biggest outbreak was recorded between February 2014 and March 2015 in the Antioquia department. Four villages (Loma Verde, Campo Alegre, Santo Domingo, and Zungo) were affected and 75 confirmed cases of *S.* Typhi were reported with an unknown source. Several different PFGE patterns were identified but COINJPPXO1.0008 (32/75; 42.6%) was the dominant pulsotype.

## Colombian Salmonella Typhi in a Latin American framework

To assess the genetic relatedness of Colombian *S.* Typhi with organisms circulating in Latin America we selected 29 pulsotypes shared by at least two organisms and compared these with

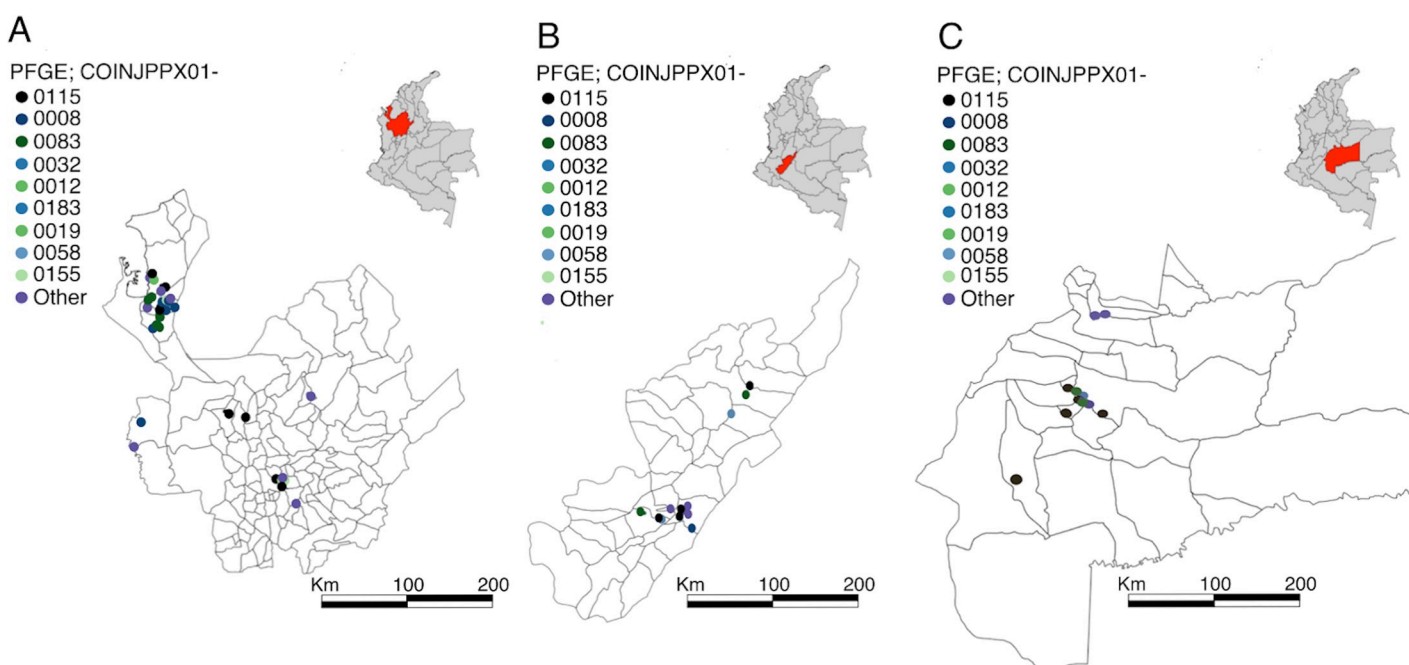

**Fig 4. The geographic distribution of typhoid fever in specific Colombian departments.** Colored dots represent the residential location and number of typhoid fever cases throughout the country by PFGE genotype (COINJPPX01-) (see key). A) Map of Antioquia department showing the location of typhoid cases by municipality; Apartadó (n = 69) and Turbo (n = 19). B) Map of Huila department showing the location of typhoid cases by municipality; Garzon (n = 15) and Agrado, Gigante, Palermo, and Neiva (all n = 5). C) Map of Meta department showing the location of typhoid cases by municipality; Granada; (n = 19), and Villavicencio, Mesetas, El Castillo, and Fuente de Oro (all n = 6). Political division maps constructed in ArcGIS 9.3 software (ESRI, Redlands, CA, USA) from polygon shapefiles accessed from https://www.diva-gis.org/. Kernel function bandwidth with a scale of outputs of 80Km.

representative restriction patterns in the regional PNLA&C database, which at the time of this study contained 967 isolates obtained from Argentina, Bolivia, Brazil, Chile, Colombia, Costa Rica, Guatemala, Paraguay, Peru, Uruguay, and Venezuela. These 967 isolates displayed 329 distinct *Xba*I restriction patterns [23]. More generally, the Latin American *S.* Typhi isolates could be broadly classified into eight regional digestion patterns (ALJPPX01.0016, ALJPPX01.0045, ALJPPX01.0048, ALJPPX01.0050, ALJPPX01.0076, ALJPPX01.0089, ALJPPX01.0191, and ALJPPX01.0195). Our analysis determined that six pulsotypes found in Colombia had also been reported in Chile, Argentina, Venezuela, and Peru. However, the three most common Colombian pulsotypes were not found among the pulsotypes reported by other Latin American countries (Fig 5).

## Discussion

This study characterized *S.* Typhi isolated from cases of typhoid fever collected by the Sivigila National Surveillance System and laboratory from across Colombia between 2012–2015. Our work demonstrates that there is a substantial burden of typhoid fever in Colombia [22]. The disease epidemiology in this Latin American country appears to be highly variable, with typhoid associated with sustained transmission in some regions and short-term outbreaks in other departments. We additionally found that typhoid is broadly distributed and caused by multiple genotypes, of which the majority may be constrained to Colombia and do not appear to circulate between other Latin American countries.

We identified a particularly high burden of typhoid in Cucutá city in Norte de Santander department and in other more remote locations including Antioquia, Meta, and Huila,

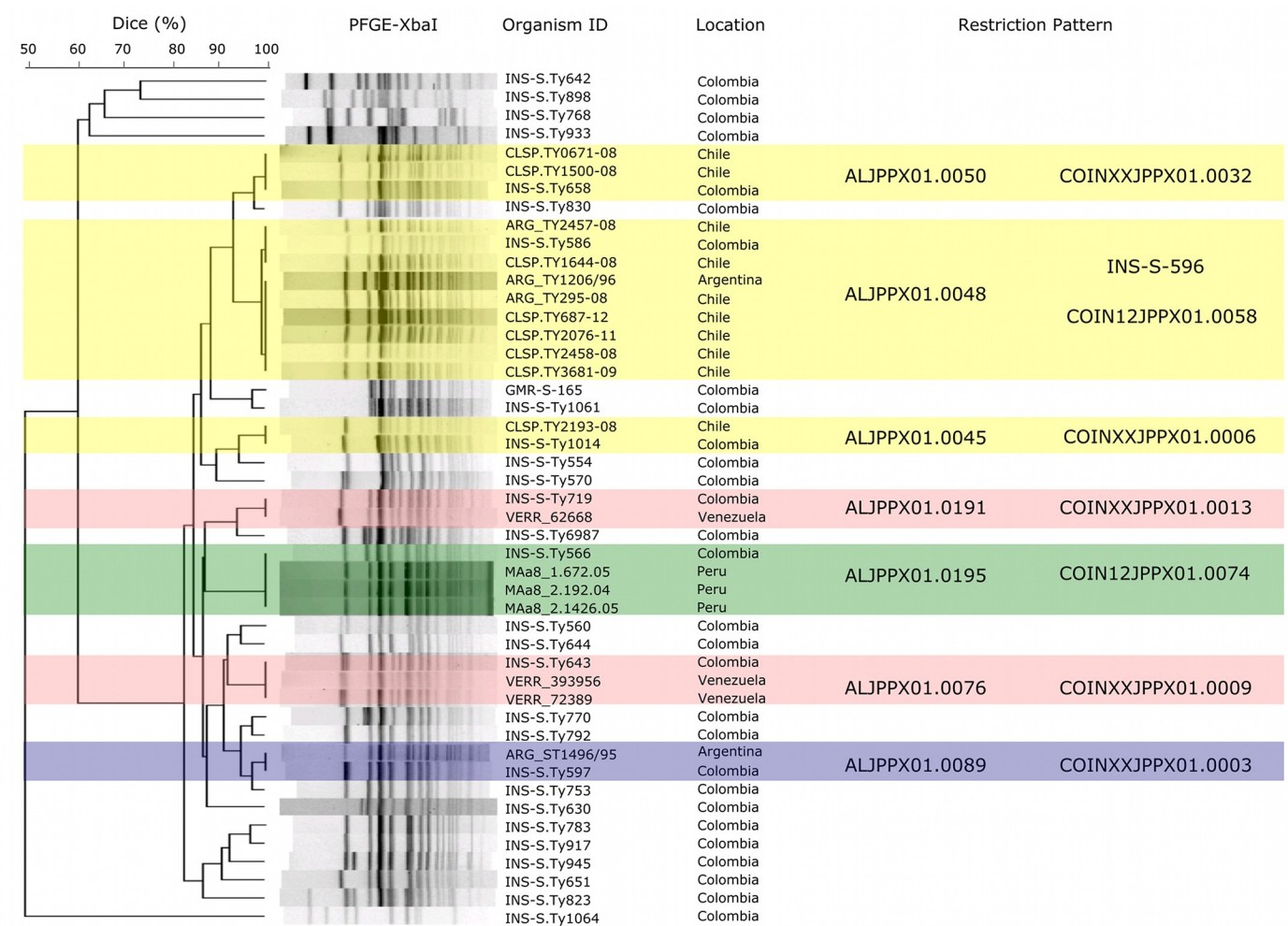

**Fig 5. Colombian *Salmonella* Typhi in a Latin American context.** Dendrogram of PFGE-*Xba*I *S*. Typhi digestions in the context of regional PFGE patterns, from the PNLA&C Database. The shared PFGE patterns between Chile ALJPPX01.0050, ALJPPX01.0045, Argentina and Chile ALJPPX01.0048, Venezuela ALJPPX01.0191, ALJPPX01.0076, Peru ALJPPX01.0195, and Argentina ALJPPX01.0089 with those found circulating in Colombia (COINXXJPPX01- and INS-S-596) are highlighted on the right of the diagram.

indicating both urban and rural disease transmission. Similar observations have been made in Sub-Saharan Africa [24]. Colombia has been classified as a country with an intermediate burden of typhoid fever (1.2 to 2.0 cases per 100,000 people/year) [4,5,25]. However, it is known that epidemics of typhoid fever can arise sporadically in Latin America [26]; therefore, we suspect the underreporting of disease in the national data capturing system as the national surveillance system is unlikely to capture the full extent of the disease burden in Colombia. Furthermore, given the geographical landscape of Colombia, there is the potential for a lack of typhoid reporting in many rural locations, leading to a shortfall in disease notification of these cases to the central surveillance system. Our data were enriched for organisms arising from the major cities in well-connected departments. This lack of an equal distribution of clinical laboratories across the country is likely to induce substantial bias in any conclusion regarding the distribution of typhoid in Colombia. Many rural locations do not have access to standardized blood culturing systems or a resident clinical microbiologist to identify infecting organisms [27]. The highest density of laboratories with the ability to perform blood culture can be found

in Nariño, Boyacá, Atlántico, Bolívar, Valle del Cauca, Santander, Antioquia, and Bogotá. Whereas, the lowest density of clinical laboratories with the ability to perform blood culture is in the departments of Vaupes, San Andres and Providence, Guainía, Quindío, Vichada, Guaviare, Amazonas, and Putumayo. Furthermore, almost half the cases were isolated as part of the national contributory health insurance scheme, suggesting bias towards those that can afford to pay for improved healthcare services. The lack of a surveillance system that encompasses the entire country, the use of non-standardized protocols, and an inconsistent health insurance system may also limit case numbers being detected and reported [28].

We observed that the majority *S.* Typhi isolates from this study were pan-susceptible to the tested antimicrobials, with MDR and fluoroquinolone resistance phenotypes being uncommon. Consequently, our data suggest that traditional first-line antimicrobials and the fluoroquinolones are likely to remain largely effective for the treatment of typhoid fever in Colombia [29]. Local treatment guidelines for typhoid do not yet exist in Colombia and use standard of care empirical treatment for a patient suspected to have typhoid fever, which is typically a third-generation cephalosporin, such as ceftriaxone. However, many hospitals follow the guidelines of the WHO and then switch to a fluoroquinolone when *S.* Typhi has been isolated from blood [30]. We identified no ESBL producing *S.* Typhi; other ESBL producing Salmonella have been reported from Latin America, but unlike the epidemiology of *S.* Typhi outside of the region, drug resistance appears not to be an issue [31,32]. This observation is probably associated with a lack of H58 (genotype 4.3.1) *S.* Typhi outside of Asia and Africa. The integration of whole genome sequencing (WGS) with conventional epidemiology has been shown to be highly valuable in the detection of outbreaks, strengthening AMR surveillance, and public health investigations [33]. Our future direction aims to introduce WGS in our current surveillance network to monitor for the emergence of H58 *S.* Typhi in Latin America and continue to track the local spread of other AMR genotypes.

This study identified specific geographical regions that may be hot spots of *S.* Typhi in Colombia. For instance, Cucutá in Norte de Santander department and two localized clusters across Antioquia (spanning Apartadó, Carepa, Chigorodo, Itsmina, Medellin, Murindo, Riosucio, Turbo, and Vigia del Fuerte) exhibited extensive case clustering of specific genotypes. These locations have a propensity for poor sanitation and are the hubs for recent massive displacement of provincial workers in a low sociodemographic population of indigenous people and immigrants [34]. The more recent typhoid epidemiology in Colombia (and other Latin American countries) may be highly volatile as a consequence of the Venezuelan migratory situation. Indeed, this highest minimum incidence was identified in Norte de Santander, which lies on the border with Venezuela. However, due to the limitations of data collection and a porous border we were only able to identify two patients that were confirmed to have entered Colombia from the state of Táchira, Venezuela. This crisis has induced a convergence of social and public health problems and should be the targeted for the provision of appropriate public health interventions and future research initiatives [35].

Previous studies have suggested that multiple *S.* Typhi PFGE patterns circulate at a regional level across Latin America [36,37]. A regional comparison between typhoid outbreaks in Argentina and Colombia found that many isolates shared highly similar restriction patterns [38]. In this study, two restriction patterns dominated, which included 36.6% of the isolates from outbreaks in the departments of Putumayo and Antioquia. More recently, we compared *S.* Typhi isolates from 2005–2008 from Argentina, Brazil, Colombia, and Chile and found that various PFGE patterns were exchanged between Colombia and Brazil [36]. Latterly, we compared *S.* Typhi isolates from Colombia with the PNLA&C database to identify shared PFGE patterns; two common Colombian restriction patterns were indistinguishable with organisms found in Argentina and Chile and were hypothesized to be associated with sustained regional

circulation [37]. A study comparing organisms isolated between 1996 and 2016 in ten countries across the continent described 278 regional *Xba*I-PFGE patterns, of which 34 were shared between several countries [23]. In this broad Latin American study, Colombia possessed 23 *Xba*I-PFGE patterns that were identical to organisms isolated in Argentina, Brazil, Chile, Guatemala, Peru, and Venezuela. These studies suggest a high level of genetic diversity of *S*. Typhi circulating within individual countries in Latin America of which some variants have the ability to spread successfully across the region. Again, the adoption of WGS should facilitate a deeper understanding of the population structure and dynamics of *S*. Typhi in this region.

This study has limitations; due to the nature of the surveillance system(s). The data may be incomplete as cases are detected passively and particular departments may have a lack of facilities for diagnosing typhoid. Additionally, the population at risk and health seeking behavior were not assessed in our study making an accurate incidence estimation unfeasible. Furthermore, bacterial genotyping was limited to PGFE, which is no longer the gold standard method for genotyping *S*. Typhi. However, due the limitation of the capacity to perform WGS across Latin America, PFGE remains the currently preferred subtyping method by PNLA&C [39]. Despite these limitations, this study highlights that there is a significant burden of typhoid in Colombia and the political instability in Venezuela may place additional pressures on typhoid control in Latin America.

Our study provides a current perspective of typhoid fever in a Latin America country and highlights the importance of pathogen-specific surveillance to add insight into the epidemiology of typhoid in this region. Sustained surveillance and the adoption of WGS in high risk areas should aid in disease control, our ability to identify new AMR variants, and permit us follow specific clones and lineages in Colombia and across Latin America.

## Supporting information

**S1 Checklist. STROBE checklist.**
(DOC)

**S1 Table. Notification of cases and annual incidence per 100,000 people of typhoid fever in Colombia by department; 2012 to 2015.**
(XLSX)

**S2 Table. Patient and isolate data.**
(XLSX)

## Acknowledgments

We express our thanks to all typhoid fever patients whose isolates were included in this project and the personnel from the local hospitals and public health laboratories in Colombia. We also thank

Instituto de Salud Pública, Santiago de Chile, Chile; Instituto Nacional de Salud, Lima, Perú and Instituto Nacional de Higiene "Rafael Rangel", Caracas, Venezuela from Pulsenet Network for allowing the regional comparison of PFGE patterns. We lastly like to thank Sandra Saavedra for her contributions and the professionals of Microbiology Group of Instituto Nacional de Salud, Bogotá, Colombia, Nancy Floréz for their collaboration in processing of the *S*. Typhi Colombian isolates and Sandra Saavedra for her collaboration.

## Author Contributions

**Conceptualization:** Paula Diaz-Guevara, Stephen Baker.

**Data curation:** Paula Diaz-Guevara, Lucy Angeline Montaño, Gabriela Zabaleta, Miriam Moroni, Josefina Campos.

**Formal analysis:** Paula Diaz-Guevara, Julio Cesar Martinez Angarita, Jaime Guerrero, Stephen Baker.

**Investigation:** Paula Diaz-Guevara, Lucy Angeline Montaño, Carolina Duarte, Gabriela Zabaleta, Julio Cesar Martinez Angarita, William León-Quevedo, Carlos Castañeda-Orjuela, Claudia Jimena Alvarez Alvarez, Jaime Guerrero.

**Methodology:** Lucy Angeline Montaño, Carolina Duarte, Gabriela Zabaleta, Julio Cesar Martinez Angarita, Duy Pham Thanh, William León-Quevedo, Carlos Castañeda-Orjuela, Claudia Jimena Alvarez Alvarez, Jaime Guerrero, Miriam Moroni, Stephen Baker.

**Project administration:** Enrique Pérez.

**Resources:** Mailis Maes, Josefina Campos, Enrique Pérez.

**Supervision:** Stephen Baker.

**Writing – original draft:** Paula Diaz-Guevara.

**Writing – review & editing:** Mailis Maes, Duy Pham Thanh, Enrique Pérez, Stephen Baker.

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
