## [Decision Letter · Decision Letter 0]

23 Sep 2019

Dear Professor Baker:

Thank you very much for submitting your manuscript "Endemic transmission of Salmonella enterica serovar Typhi in Colombia, 2012-2015" (#PNTD-D-19-01222) for review by PLOS Neglected Tropical Diseases. Your manuscript was fully evaluated at the editorial level and by independent peer reviewers. The reviewers appreciated the attention to an important problem, but raised some substantial concerns about the manuscript as it currently stands. These issues must be addressed before we would be willing to consider a revised version of your study. We cannot, of course, promise publication at that time.

We therefore ask you to modify the manuscript according to the review recommendations before we can consider your manuscript for acceptance. Your revisions should address the specific points made by each reviewer. 

When you are ready to resubmit, please be prepared to upload the following:

(1) A letter containing a detailed list of your responses to the review comments and a description of the changes you have made in the manuscript.

(2) Two versions of the manuscript: one with either highlights or tracked changes denoting where the text has been changed (uploaded as a "Revised Article with Changes Highlighted" file); the other a clean version (uploaded as the article file).

(3) If available, a striking still image (a new image if one is available or an existing one from within your manuscript). If your manuscript is accepted for publication, this image may be featured on our website. Images should ideally be high resolution, eye-catching, single panel images; where one is available, please use 'add file' at the time of resubmission and select 'striking image' as the file type. 

Please provide a short caption, including credits, uploaded as a separate "Other" file. If your image is from someone other than yourself, please ensure that the artist has read and agreed to the terms and conditions of the Creative Commons Attribution License at http://journals.plos.org/plosntds/s/content-license (NOTE: we cannot publish copyrighted images). 

(4) If applicable, we encourage you to add a list of accession numbers/ID numbers for genes and proteins mentioned in the text (these should be listed as a paragraph at the end of the manuscript). You can supply accession numbers for any database, so long as the database is publicly accessible and stable. Examples include LocusLink and SwissProt.

(5) To enhance the reproducibility of your results, we recommend that you deposit your laboratory protocols in protocols.io, where a protocol can be assigned its own identifier (DOI) such that it can be cited independently in the future. For instructions see http://journals.plos.org/plosntds/s/submission-guidelines#loc-methods

While revising your submission, please upload your figure files to the Preflight Analysis and Conversion Engine (PACE) digital diagnostic tool, https://pacev2.apexcovantage.com/ PACE helps ensure that figures meet PLOS requirements. To use PACE, you must first register as a user. Then, login and navigate to the UPLOAD tab, where you will find detailed instructions on how to use the tool. If you encounter any issues or have any questions when using PACE, please email us at figures@plos.org.

We hope to receive your revised manuscript by Nov 22 2019 11:59PM. If you anticipate any delay in its return, we ask that you let us know the expected resubmission date by replying to this email.

To submit a revision, go to https://www.editorialmanager.com/pntd/ and log in as an Author. You will see a menu item call Submission Needing Revision. You will find your submission record there. 

Sincerely,

Andrew S. Azman

Deputy Editor

Reviewer's Responses to Questions

**Key Review Criteria Required for Acceptance?**

**Methods**

-Are the objectives of the study clearly articulated with a clear testable hypothesis stated?

-Is the study design appropriate to address the stated objectives?

-Is the population clearly described and appropriate for the hypothesis being tested?

-Is the sample size sufficient to ensure adequate power to address the hypothesis being tested?

-Were correct statistical analysis used to support conclusions?

-Are there concerns about ethical or regulatory requirements being met?

Reviewer #1: The objectives of this study are clear, and the study design is appropriate. The population studied is described and the sample size large enough to draw conclusions. The statistical analysis appears appropriate and I have no ethical concerns.

Reviewer #2: Methods 

1. Line 130. It is mentioned that certain areas of the country have strengthened their surveillance for typhoid fever. If those specific areas or departments can be cited that would be helpful.

2. Line 133: Defining “non-endemic” as “incidences lower than the national average” is not the usual usage and meaning of “endemicity”. Endemicity repetitive, predictable occurrence over time rather than relative incidence (incidence against a national average). Characterizing all regions below the national average as non-endemic could lead to under-reporting of endemicity and over-reporting of sporadic transmission. The U.S. CDC and W.H.O / P.A.H.O. have definitions of sporadic, endemic, hyperendemic transmission; as well as epidemic, outbreak, cluster, pandemic, etc.

**Results**

-Does the analysis presented match the analysis plan?

-Are the results clearly and completely presented?

-Are the figures (Tables, Images) of sufficient quality for clarity?

Reviewer #1: The analysis matches the plan and the results are clear and clearly presented. I found the maps to be a little blurred.

Reviewer #2: 1. Of 468 S. Typhi isolates, 402 came from seven departments. The department of Norte de Santander accounted for 190 (47.3%) of these 402 isolates. This result immediately catches the attention of the reader.

2. Line 193-194, the statement “almost half of cases were affiliated to the national contributory health insurance scheme” is explained. If the authors are implying that this impacts surveillance, they should elaborate on this point. Also, “insurance scheme” is not discussed elsewhere, except itemized under Table 1. 

3. Figure 1 should explain the quantitative significance of the dot size in the legend and/or caption. If the dot sizes have no absolute meaning and are listed in steadily decreasing order based on PFGE type prevalence, perhaps to assist in differentiation by both size and color, then that must be stated so that misinterpretation is avoided. Currently, it might be interpreted that one dot of -0115 represents a prevalence 25-times greater than one dot of -0155 (roughly gauging relative dot size). Also, does each dot represent a hospital/reporting site or a patient’s home coordinates?

4. Lines 205-206. Does Sivigila define “endemic”? 

5. The first paragraphs of the Results discuss the number of cases by department and smaller administrative units. A table that presents: the mean annual population of each of these seven departments during the years 2012-2015 (derived from either the 2005 or 2018 census data); the mean number of S. Typhi cases over 2012-2015 and the mean annual incidence per 100,000 persons for those years, would be very helpful. Some departments, such as Norte de Santander, that have been the site of extensive legal and illegal immigration from Venezuela during 2014 and 2015 may have notable numbers of high-risk persons who are not reflected in the 2018 census. Nevertheless, whatever denominators are available should be used to calculate incidence rates. Then some of the figures can be modified to have spot maps based on incidence rather than just number of cases, should be crafted. 

6. Line 234: Table 3 does not represent the result that is stated immediately prior and in reference to the table. Please check this.

7. Line 244-245: The sentence, “Notable, Group III contained the majority of organisms […] and nalidixic acid (n=10)” is misleading because Group III also contained 356 out of 402 characterized isolates. A percentage/proportion might be included in parentheses and a statistical test could be performed to support this association, if it is to be deemed “notable.” It should also be further discussed in the discussion section.

8. Figure 2: The four maps are inconsistent with the text (lines 236-247) and the figure 2 caption. See below. 

Lines 236-247: Group 1 has 13 isolates in 6 departments. Group 2 has 27 isolates in 6 departments. Group 3 has 356 isolates in 14 departments. Group 4 has 6 isolates in 4 departments.

Figure 2 Caption: Group 1, purple, in 7 departments (should it be 6?). Group 2, yellow, in 6 departments. Group 3, orange (should it be green?), in 5 departments (should it be 14?). Group 4, green (should it be orange?), in 14 departments (should it be 4?).

There are several inconsistencies (numbers and colors) that need addressing. 

The data contained within lines 265-274 could be better understood and discussed in a table or figure. 

Lines 287-288: What are the “three most common Colombian pulsotypes” specifically according to Figure 5? They are not clearly labeled or identified, and the labels in Figure 5 do not match the labels in Figure 1 or Table 3. 

9. It was not possible to download reference #22. Please check all citations and links carefully.

10. Line 197: “liquid corporal-secretions” should be changed to “normally sterile body fluids”. 

11. Figure 1 caption – “Kerner” seems to be a misspelling of “Kernel”.

12. Figure 3 requires a scale on the map and again, “Kerner” seems to be a misspelling of “Kernel”.

13. Figure 4 caption: Lists cases by municipality, but the maps in Figure 4 do not label these municipalities. 

14. Lines 276-288 “Colombian Salmonella Typhi in a Latin America framework”. In order to assess the relevance of this analysis the readers need to know how representative the isolates are for the other countries and how these isolates were obtained. Without this information it is hard to judge the relevance of comparing a few isolates from other countries in South America.

**Conclusions**

-Are the conclusions supported by the data presented?

-Are the limitations of analysis clearly described?

-Do the authors discuss how these data can be helpful to advance our understanding of the topic under study?

-Is public health relevance addressed?

Reviewer #1: The conclusions are supported by the data and the limitations explored. There is discussion of how the data could be used and the public health implications.

Reviewer #2: 1. Line 300. “We identified a particularly high burden of typhoid in Cucutá”, the capital city of Norte de Santander department. This city has been an epicenter of Venezuelan migration. The isolates from Cucutá should be compared to Venezuelan isolates of S. Typhi, if the authors have some. 

2. Do the authors have any way to differentiate which isolates came from Venezuelan immigrants and refugees? Minus the isolates from this city and this department, the total Colombia burden drops almost in half. So it is critical to attempt to ascertain how many of the typhoid cases derive from the Venezuelan refugee crisis.

3. Lines 317 – 330. This is an important and well written paragraph. 

4. Lines 332 – 337. This paragraph addresses in detail the issues raised above with respect to the status of Venezuelan refugees in Cucutá and elsewhere in Norte de Santander.

5. Lines 341-358. To reiterate, without knowing more about the isolates from the other Latin American countries, including how they were selected, it is difficult for the reader to accept categorically the conclusions drawn by the authors. More information about these strains and softening of statements would strengthen the paper.

**Editorial and Data Presentation Modifications?**

Reviewer #1: Line 126: sentence does not make sense. Do you mean ’Cases were defined as those with …’?

Line 131: Not sure what reiterative epidemiological behaviour is. Could this be clearer?

Line 197: What are liquid corporal-secretions?

Line 203 and line 205: How does the public health department define outbreaks, endemic and non-endemic?

Line 306: I could not see these discrepancies discussed in the results but may have missed it

Line 319: What are the recommended first line antbiotics for typhoid? Are there national guidelines?

Line 320: I could not see any results that confirmed that the extended spectrum cephalosporin resistant strain had an ESBL phenotype. It looks as if this strain was negative by PCR for blaSHV, blaTEM, and blaCTX-M?

Table 1

Please explain some of the variables (perhaps as a footnote): Regime type; Categories of health coverage; municipal centre and populated centre.

Reviewer #2: In the spirit of being helpful, the following typographical and grammatical edits are offered:

• Line 56: A period is missing at the end of the first sentence

• Line 59: “we aimed to add some insight” is informal

• Line 62: “widely is” should be changed to “is widely”

• Line 63: “major” should be written as “majority”

• Line 64: “epidemiology of typhoid of Colombia” should be rewritten to “epidemiology of typhoid in Colombia” or similar

• Line 64: “distinct to” should be written as “distinct from”

• Line 65: “organisms that circulating nationally” should be written as “organisms that are circulating nationally” or “organisms that circulate nationally”

• Line 83: misplaced comma between “disease, which”

• Line 302: Run-on sentence between “transmission, similar”

• Line 303: Missing articles; “currently classified as country with intermediate burden” should include “a” so it reads “as a country” and “an” so it reads “with an intermediate burden”

• Line 341: Change 5th word “the” to “that”

• Line 360: run-on sentence between “system(s)” and “the data”

**Summary and General Comments**

Reviewer #1: Typhoid fever was a major public health problem in Latin America in past decades but in recent years it has appears to have declined as a problem. This report is a useful reminder that for some countries and populations it remains an important disease. Importantly, unlike other regions of the world, antimicrobial resistance was uncommon and restricted to low levels of resistance to ampicillin and fluoroquinolones. A diversity of genetic types was described and limited overlap with genetic types in other Latin American countries. There was some evidence of areas with a higher incidence.

Detection of cases is dependent on a positive blood or bone marrow culture. It would be helpful to have some more detailed background information about the distribution of laboratories with the capacity for blood culture. Would most patients with a fever been investigated with a blood culture? In particular would blood cultures be performed in children and in smaller hospitals outside the main department centres or is it possible that there is under-ascertainment of cases in these groups?

Reviewer #2: The manuscript reports that over a 4-year period (2012-2015) in Colombia the national public health surveillance protocol for the identification of typhoid and paratyphoid fever yielded 468 confirmed cases of typhoid fever that had an archived Salmonella Typhi isolate. Notably, 420 of the 468 isolates originated from seven departments of Colombia, with 190/420 isolates coming from a single department, Norte de Santander, which borders Venezuela. Most cases occurred in the main metropolitan centers in the affected departments and 68.2% of aces were in persons > 15 years of age. Importantly, the vast majority of the isolates were sensitive to first-line antibiotics and none were H58 lineage. Pulse field gel electrophoresis (PFGE) was performed on 402 S. Typhi isolates allowing special distribution analyses to be carried out. These data shed light on some geographic areas and sub-populations in Colombia where typhoid fever remains a public health problem. These data do provide helpful information on typhoid fever in Colombia. Whilst the PFGE patterns if Colombian isolates are useful for analyzing relatedness among strains from within the same department and between different departments of Colombia, the attempt to relate the PFGE patterns to isolates from other Latin American countries was not particularly helpful because the source of the strains from other countries was not described in any detail. 

Need a table that shows population, cases and incidence per 100,000 for the seven departments and for Cucutá”, the capital city of Norte de Santander department. This city has been an epicenter of Venezuelan migration. The isolates from Cucutá should be compared to Venezuelan isolates of S. Typhi, if the authors have some.

PLOS authors have the option to publish the peer review history of their article (what does this mean?). If published, this will include your full peer review and any attached files.

Reviewer #1: No

Reviewer #2: No

---

## [Editor Report · Decision Letter 1]

9 Jan 2020

Dear Professor Baker,

We are pleased to inform you that your manuscript, "Surveillance of Salmonella enterica serovar Typhi in Colombia, 2012-2015", has been editorially accepted for publication at PLOS Neglected Tropical Diseases.

Before your manuscript can be formally accepted and sent to production you will need to complete our formatting changes, which you will receive in a follow up email. Please note: your manuscript will not be scheduled for publication until you have made the required changes.

IMPORTANT NOTES

* Copyediting and Author Proofs: To ensure prompt publication, your manuscript will NOT be subject to detailed copyediting and you will NOT receive a typeset proof for review. The corresponding author will have one final opportunity to correct any errors when sent the requests mentioned above. Please review this version of your manuscript for any errors.

* If you or your institution will be preparing press materials for this manuscript, please inform our press team in advance at plosntds@plos.org. If you need to know your paper's publication date for media purposes, you must coordinate with our press team, and your manuscript will remain under a strict press embargo until the publication date and time. PLOS NTDs may choose to issue a press release for your article. If there is anything that the journal should know, please get in touch.

*Now that your manuscript has been provisionally accepted, please log into EM and update your profile. Go to http://www.editorialmanager.com/pntd, log in, and click on the "Update My Information" link at the top of the page. Please update your user information to ensure an efficient production and billing process.

*Note to LaTeX users only - Our staff will ask you to upload a TEX file in addition to the PDF before the paper can be sent to typesetting, so please carefully review our Latex Guidelines [http://www.plosntds.org/static/latexGuidelines.action] in the meantime.

Best regards,

Andrew S. Azman

Deputy Editor

Andrew Azman

Deputy Editor

---

## [Editor Report · Acceptance letter]

25 Feb 2020

Dear Professor Baker,

We are delighted to inform you that your manuscript, "Surveillance of *Salmonella enterica serovar*  Typhi in Colombia, 2012-2015," has been formally accepted for publication in PLOS Neglected Tropical Diseases.

Best regards,

Serap Aksoy

Editor-in-Chief

Shaden Kamhawi

Editor-in-Chief
